# Computer-based inhibitory control training in children with Attention-Deficit/Hyperactivity Disorder (ADHD): Evidence for behavioral and neural impact

**Kristin N. Meyer**[1], **Rosario Santillana**[2], **Brian Miller**[3], **Wes Clapp**[3], **Marcus Way**[4], **Katrina Bridgman-Goines**[5], **Margaret A. Sheridan**[1,6]*

1 Department of Psychology and Neuroscience, University of North Carolina, Chapel Hill, NC, United States of America, 2 Davis Medical School, University of California, Sacramento, California, United States of America, 3 NeuroScouting, LLC, Cambridge, MA, United States of America, 4 Amazon, Boston, MA, United States of America, 5 Emory University, Atlanta, GA, United States of America, 6 Department of Developmental Medicine, Boston Children's Hospital, Boston, MA, United States of America

* Sheridan.margaret@unc.edu

**Data Availability Statement:** All relevant data are within the manuscript and its Supporting Information files.

## Abstract

Attention-deficit hyperactivity disorder (ADHD) is the most commonly diagnosed psychological disorder of childhood. Medication and cognitive behavioral therapy are effective treatments for many children; however, adherence to medication and therapy regimens is low. Thus, identifying effective adjunct treatments is imperative. Previous studies exploring computerized training programs as supplementary treatments have targeted working memory or attention. However, many lines of research suggest inhibitory control (IC) plays a central role in ADHD pathophysiology, which makes IC a potential intervention target. In this randomized control trial (NCT03363568), we target IC using a modified stop-signal task (SST) training designed by NeuroScouting, LLC in 40 children with ADHD, aged 8 to 11 years. Children were randomly assigned to adaptive treatment (n = 20) or non-adaptive control (n = 20) with identical stimuli and task goals. Children trained at home for at least 5 days a week (about 15m/day) for 4-weeks. Relative to the control group, the treatment group showed decreased relative theta power in resting EEG and trending improvements in parent ratings of attention (i.e. decreases in inattentive behaviors). Both groups showed improved SST performance. There was not evidence for treatment effects on hyperactivity or teacher ratings of symptoms. Results suggest training IC alone has potential to positively impact symptoms of ADHD and provide evidence for neural underpinnings of this impact (change in theta power; change in N200 latency). This shows promising initial results for the use of computerized training of IC in children with ADHD as a potential adjunct treatment option for children with ADHD.

**Funding:** This study was supported by National Institute of Mental Health (NIMH) in the form of grants awarded to MS (K01MH092555) and BM and WC (R43 MH095282), Robert Wood Johnson Foundation in the form of a grant awarded to MS (Cohort 5, Health and Society Scholars at Harvard), and National Science Foundation Graduate Research Fellowships Program in the form of a grant awarded to KM (DGE-1650116). Neuroscouting, LLC. also provided support in the form of salaries for BM and WC. The specific roles of these authors are articulated in the 'author contributions' section. The funders had no role in study design, data collection and analysis, decision to publish, or preparation of the manuscript.

**Competing interests:** The authors have read the journal's policy and have the following competing interests: BM and WC own Neuroscouting, LLC. This does not alter our adherence to PLOS ONE policies on sharing data and materials. There are no patents, products in development or marketed products associated with this research to declare.

## Introduction

Attention-deficit/hyperactivity disorder (ADHD) is the most commonly diagnosed psychological disorder in childhood [1], and it confers significant risk for poor outcomes in adolescent and adult mental health, economic success, and school performance [2–5]. While medications effectively reduce ADHD symptoms, medication adherence is often low in this population, with non-adherence as high as 13–64%, likely due to the associated stigma, cost, or safety concerns [6, 7]. As such, exploring non-pharmacological interventions that work in tandem with existing treatment is necessary.

There has been a recent explosion in studies aimed at improving cognitive function through computerized training [8, 9], in part because this form of intervention has potential a successful adjunct treatment for individuals with psychopathology. ADHD is a likely target for such computerized training interventions because the primary deficit in ADHD is generally cognitive in nature [10, 11]. A large body of research suggests that children with ADHD show deficits in executive functioning [12, 13]. As such, many cognitive training programs have targeted executive functioning in ADHD with mixed success, as reported in more detail below. Executive functioning describes a set of effortful processes that regulate cognitive resources and actions in service of higher-order goals [14]. The major components of executive functioning include updating working memory, inhibitory control (IC) over prepotent responses, and shifting between mental sets [14, 15]. While ADHD is associated with deficits across executive functioning domains, decades of research has identified IC as a central deficit in ADHD [11, 12, 16].

IC abnormalities in children with ADHD are evidenced both in behavioral performance as well as in the structure and function of neural systems that support IC function [17–24]. Related to this IC deficit, children with ADHD have increased power in theta frequencies as measured by EEG during rest [25, 26], disrupted and/or inefficient recruitment of the prefrontal cortex during IC task performance [18], and decreased amplitude and increased latency for an ERP component associated with successful inhibition (N200) compared to children without ADHD [27–30]. The N200 is an ERP component that reaches peak amplitude at about 200 ms after stimulus presentation and has been associated with IC [31, 32]. Given the robust evidence for dysfunction in this particular cognitive and neural system, computerized training of IC shows promise as a potential adjunct treatment for children with ADHD. Previous studies have instead typically targeted other areas of cognitive function including attention, working memory, or a combination of executive functions, with mixed success as we describe below.

Initial cognitive training efforts in children with ADHD focused on training attention directly [33]. Cognitive training programs typically assess training effects on both near transfer (i.e. intervention effects on tasks in the trained domain) and far transfer (i.e. intervention effects on untrained skills). The initial non-computerized attention training programs, Attention Process Training (APT) and *Pay Attention*! showed near transfer improvements in non-trained attention tasks [34–39]. Later attention training programs, such as AixTent, were computerized and adaptive (in which task difficulty depends on performance). These programs also produced near transfer improvements as compared to visual perception training but no improvements in parent or teacher reports of symptoms [40, 41].

More recent efforts have targeted working memory training, including CogMed. This training program was also adaptive in nature, for which task difficulty increased based on participant performance. It is thought that the increasing difficulty of adaptive training does not allow for reliance on strategies specific to the task or habitual processing, which means that participants must engage executive functioning processes to complete the tasks [42]. And within working memory training, adaptive training tends to yield more far transfer effects

than non-adaptive training, while non-adaptive training typically only exhibits near transfer effects [43]. An initial study comparing adaptive and non-adaptive working memory training tasks found adaptive training resulted in improvements on a range of non-trained cognitive tasks (including working memory, complex reasoning, and response inhibition) as well as reductions in parent-rated ADHD symptoms [44]. While the results of this initial study were incredibly promising, subsequent studies using the CogMed training program have failed to replicate the impact of training on ADHD symptoms [45–47]. One study specifically elected to implement a working memory training that also had an inhibitory control component, the n-back task, and found that this training impacted performance on untrained working memory and inhibitory control tasks with marginal impact on parent reported ADHD symptoms [48].

While some studies have found improvements from cognitive training, there is still debate regarding the efficacy of cognitive training broadly. For one, it has also been suggested due to lack of well-controlled comparisons in cognitive training studies, placebo effects may be driving some results [49]. We aim to account for in the current study by utilizing the same training games across control and treatment groups with the specific manipulation an adaptive version for the treatment group and non-adaptive version for the control group. Furthermore, there are mixed findings regarding degree to which cognitive training can have meaningful impact on functioning beyond improvement in a specific task, as findings from recent meta-analyses have been mixed [50–52]. One meta-analysis discovered that both the effect size and heterogeneity of the effect size of cognitive training on improvements differed depending on the population receiving the training [53], suggesting some groups may benefit more than others. Meta-analyses of cognitive training in children with ADHD have typically reported small effects that are limited to near transfer improvements [33, 54, 55]; however, large effects on ADHD symptoms are observed when multiple neuropsychological processes are targeted [54]. Of the studies reported to target multiple processes, most included an IC component [54]. In the current study, we address whether targeting IC alone in cognitive training can reduce ADHD symptoms.

To date, the most successful interventions in individuals with ADHD are those that have implemented working memory or attention training along with IC training. In some studies comparing adaptive and non-adaptive versions of a training program that used a variety of tasks, including tasks with IC components, the adaptive training group showed far transfer improvements in both parent and non-parent ratings of inattention and hyperactivity, which were maintained for as long as 9 weeks [56–58]. Another program primarily targeted training aspects of attention, but also included training on a sustained attention task with an IC component (the Computerized Continuous Performance Task), and found that training led to significant reductions in parent-reported inattention symptoms [59]. In at least one of these studies, symptom reduction was accompanied by differences in neural function previously associated with ADHD diagnosis, including reduced relative theta power as measured by EEG [57]. Another study that trained processing speed, attention, working memory, cognitive flexibility, and IC in children with ADHD found that training altered neural activity during sustained attention and working memory tasks; however, no changes in neural activity during the IC task was observed [60]. These 'mixed' executive function trainings which added IC training to existing tasks appeared to be more successful at eliciting symptom reduction and impacting neural function than training focused only on working memory.

Given these findings, and previous work demonstrating that IC is a central deficit in ADHD, we investigated a training program that specifically targeted IC. While previous evidence indicates that training working memory alone does not consistently shift symptoms of ADHD, we hypothesize that training IC alone would reduce ADHD symptoms. Further, we measured potential neural mechanisms through which training IC might function. We expect to extend the previous finding indicating that combined IC and working memory training

reduces theta oscillatory activity in children with ADHD to show that these reductions can occur from IC training alone. Given previous data documenting that children with ADHD have smaller and longer latency N200 components during successful inhibition on stop signal tasks (SST), we expected that this component might be impacted by using an SST to train IC. However, since no prior study had examined the N200 following computerized training, we did not have an explicit hypothesis with regards this effect. Thus, exploratory analyses will test whether IC training impacts N200 amplitude or latency during inhibition on an SST. Information concerning these hypotheses would contribute to making training regimens for children with ADHD efficient by identifying the primary cognitive skill that requires intervention and by identifying potential neural pathways affected by cognitive training. Here we report on a randomized control study to test specific effects of IC training on ADHD symptoms and neural activity in children with ADHD by using an adaptive IC training for the treatment group and non-adaptive IC training for an active control group.

## Methods and materials

### Participants and procedure

Participants were recruited from the Participant Registry Database of Boston Children's Hospital, a list of families interested in research in the greater Boston area. Participant recruitment began July 18th, 2013 and all follow-up was completed March 3, 2014. Parents of participants provided written informed consent and children provided assent. Verbal consent was obtained from parents and children to contact the child's teacher (N = 37) or a non-parental adult (N = 4) that had cared for them (e.g. regular babysitter). Children were between 8 and 11 years of age and had been diagnosed with ADHD by a licensed clinician and had consistent access to Wi-Fi at home. Participants were excluded for any known genetic abnormalities, a diagnosis of autism spectrum disorder, or current use of medication for psychiatric disorders other than ADHD (e.g. SSRIs). Initially 41 families agreed to participate, and 1 family discontinued in the first week due to child dissatisfaction with the games. The final sample was 40 participants (12 female, 10.33 ± 1.46 years) with an ADHD diagnosis (Fig 1). Participants were randomized to either a treatment (n = 20) or control (n = 20) condition (see Table 1 for baseline characteristics). Both participants and parents were blind to study condition. Randomization was conducted by alternating between treatment and control group assignment between participants. The control group was significantly older than the treatment group ($p = .03$), however, the groups did not differ on gender, externalizing scores, or internalizing scores (all $p$'s > .2). Confirmation of ADHD diagnosis was performed using the DISC-IV. No difference in percent confirmed positive ADHD diagnosis was observed between treatment and control groups.

Within the whole sample, 36 children were taking medications (33 stimulant, 3 non-stimulant). All individuals taking stimulant medication stopped stimulant use at least 24 hours prior to each lab visit, confirmed by parent report on the day of the visit. Thus, the study design consisted of a pre-training assessment session (baseline) conducted off stimulant medication, 4 weeks (20 sessions) of computerized training on their medication as usual, and a post-training assessment session (endline) collected off stimulant medication. Resting state EEG data, ERPs in response to an SST, and parent and teacher report of ADHD symptoms were assessed before and after training. The Boston Children's Hospital Institutional Review Board approved all study protocols (Approval Date: November 6, 2013).

The study is registered on ClinicalTrials.gov (NCT03363568). This study was registered after data collection, because the data was collected prior to the NIH revising the definition of a clinical trial in 2014. The authors confirm that all ongoing and related trials for this intervention are now registered.

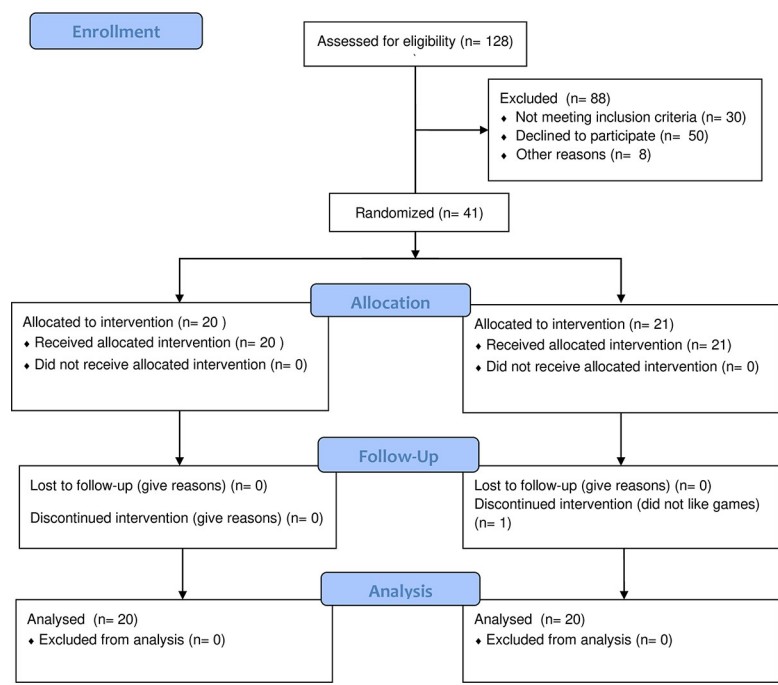

**Fig 1. CONSORT Participant flow diagram.**

## Confirmation of ADHD diagnosis

Diagnosis was confirmed at the first visit using the ADHD module of the diagnostic interview schedule for children (DISC-IV) [61] and the child behavior checklist (CBCL) [62]. Of the 40 participants, 37 children met criteria or subthreshold criteria for ADHD on the DISC-IV and/ or had a T-score above 65 on the ADHD subscale of the CBCL. Three participants (2 controls and 1 treatment) did not meet criteria on the DISC-IV but did have T-scores on the CBCL of

**Table 1. Baseline demographic and diagnostic information.**

|  | Treatment Group | Control Group | *P* |
|---|---|---|---|
| **Age in years** | 9.84 ± 1.73 | 10.82 ± 0.93 | .03 |
| **Gender (Percent female)** | 35% | 25% | .50 |
| **DISC-IV ADHD (Percent positive diagnosis)** | 70% | 80% | .48 |
| **CBCL Externalizing (T-score)** | 56.95 ± 9.7 | 58.00 ± 8.7 | .72 |
| **CBCL Internalizing (T-Score)** | 58.15 ± 9.9 | 57.80 ± 12.4 | .92 |
| **CBCL ADHD (Percentile)** | 92.05 ± 7.22 | 90.40 ± 11.55 | .57 |

Reported *p*-values obtained from independent samples t-test.

*Notes*: DISC-IV = diagnostic interview schedule for children [61]. CBCL = child behavior checklist [62].

60 or higher. All analyses were run with and without these participants and all results were unchanged. Of those who met criteria on the DISC-IV, most met criteria for the Inattentive (Treatment Group: 57%; Control Group: 56%) or Combined (Treatment: 36%; Control: 38%) subtypes with relatively few meeting criteria for the Hyperactive subtype (Treatment: 7%; Control: 6%). This is consistent with findings that the Inattentive and Combined subtypes of ADHD are the most common in children of this age [63].

## ADHD symptoms

At pre-training and post-training assessment sessions, parents completed the Swanson, Nolan, and Pelham-IV Questionnaire (SNAP-IV; [64]) and the Conner's Parent Rating Scale [65], each of which includes hyperactivity adn inattention subscales. Teacher reported scores of inattention and hyperactivity were obtained with Conner's Teacher Rating Scale administered within one week of pre-training and post-training sessions [66].

## Measures and tasks

**Electroencephalogram (EEG) recording.**   Data used for both resting state EEG and ERP analyses was acquired using a 128 HydroCel Sensor Net System (EGI, Inc., Eugene, OR; http://www.egi.com) with an Ag/AgCl- coated, carbon-filled plastic electrode and sponge. EEG data were amplified using NetAmps 200 amplifiers and recorded on a nearby computer using Net-Station 4.3.1 (Electrical Geodesics Inc., Eugene, OR). Data were sampled at 250 Hz with a bandpass filter of 0.1 to 100 Hz. Recorded EEG was digitized with a 12-bit National Instruments Board (National Instruments Corp., Woburn MA) and referenced on-line to a single vertex electrode (Cz).

**EEG acquisition and analysis.**   For resting-state EEG recording, participants alternated between sitting with eyes open and eyes closed for 7 trials of 30 seconds each [67]. This approach was chosen to maximize the amount of data collected without artifact in children with ADHD. In total, 3.5 minutes of EEG data were collected for both eyes open and eyes closed.

Data pre-processing was performed using NetStation tools and implemented in NetStation 4.3.1. Using artifact detection, all segments (of 30 seconds) with eye-blinks, eye-movements, or bad channels were identified. Channels were identified as bad if they contained high frequency noise or voltage difference greater than +/- 80 uv. Entire segments were excluded from further analysis if more than 15% of electrodes were excluded from that segment. Next, for each segment, a trained research assistant reviewed each segment and excluded electrodes from a given 30 second segment if they contained artifacts for more than half of that segment—these channels were replaced with an average of its nearest neighbors using bad channel replacement algorithms available in Net Station. At least of 2 minutes (4 segments) of useable data for eyes open and eyes closed conditions was required for a participant's data to be included in the final analyses. Following the exclusion of specific channels and segments, all electrodes around the outside of the scalp, which are known to exhibit the most noise, were excluded from further analysis (electrodes 8, 14, 21, 25, 48, 49, 56, 63, 68, 73, 81, 88, 94, 99, 107, 113, 119, 125, 126, 127, 128). After excluding these electrodes, data was re-referenced from Cz to the average signal. Following these pre-processing steps, data was exported as a concatenated time series into EEGLAB (http://sccn.ucsd.edu/eeglab). Each time series was again visually inspected by a trained research assistant, who further excluded instances of eye blinks, saccadic eye movements, and non-specific movement artifacts that had not been identified using the automatic algorithms in NetStation. On average about 19.65 seconds of data was excluded using visual inspection in EEGLAB.

Of the original 40, 34 children were included in the final EEG analysis, 6 children were excluded (5 treatment, 1 control) due to excessive movement, artifact, or technical difficulties in data acquisition at either the pre- or post- training session. For pre-treatment, an average of 1.60 minutes of eyes open data and 1.68 minutes of eyes closed data in the treatment condition and an average of 1.65 minutes of eyes open data and 1.73 minutes of eyes closed data in the control condition was included in the final analysis. The average amount of data included in final analyses from post-treatment was 1.65 minutes of eyes open data and 1.69 minutes of eyes closed data in the treatment condition and 1.67 minutes of eyes open data and 1.71 minutes of eyes closed data in the control condition. These averages did not differ between treatment and control (all $p$'s > .3).

For each participant, relative power in the theta band (5–8 Hz) was calculated for four regions of interest analogous to bilateral parietal and frontal electrodes in the international 10–10 system (Fig 2) using frequency band analyses implemented in EEGLAB. Power spectral densities were calculated in accordance with Welch's [68] method using a 1024-point (4.096s) Hanning window and averaging with 50% overlapping segments. Prior to frequency band analysis, each power spectral density was multiplied by the frequency resolution (0.244 Hz) to convert it to a power spectrum (rather than a power spectral density). Power spectra for each region of interest were calculated by averaging together the power spectra of the three individual electrodes within the given region. In order to control for differences in absolute amplitude of the EEG signal, which can be highly variable and subject to various sources of noise, we report on relative power. The relative power was calculated by dividing the power within the theta band by the total power across all of the bands.

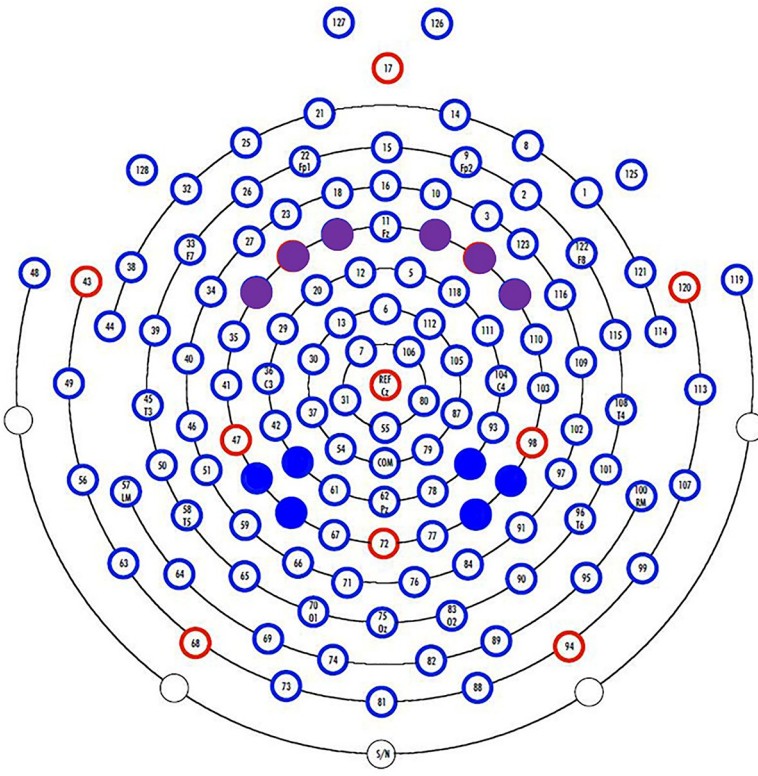

**Fig 2. EEG electrode map.** Regions of interest were bilateral parietal (solid blue) and bilateral frontal electrodes (solid purple). Electrodes were selected using the international 10–10 system.

**Event-related potential (ERP).** Event-related potentials (ERPs) were recorded, as described above, while each child completed a modified SST identical to the *Baseball Game*, described above (and see Fig 3), used for training except that children pressed a spacebar to hit the ball instead of tapping the iPad. During ERP acquisition, participants sat in front of a computer, in a quiet room by themselves and played the SST game at baseline and endline. The SST lasted approximately 7 minutes and included 60 stop trials. Prior to training there were no

A.

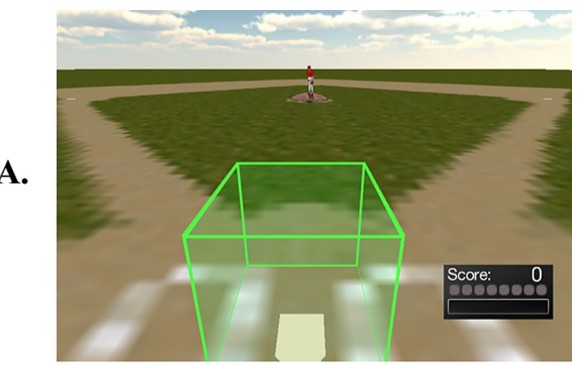

B.

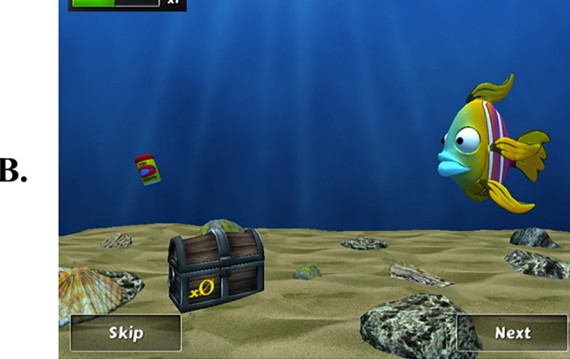

C.

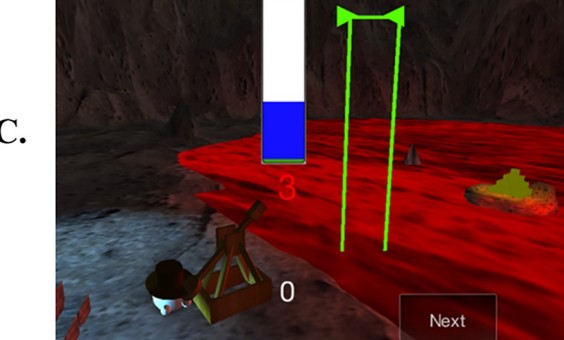

**Fig 3.** Training program games created by NeuroScouting, LLC: (a) Baseball game. (b) Feed the fish game. (c) Catapult game.

significant differences in task performance, measured as point of subjective equality (PSE; described below) and errors of omission, between children in the treatment and control groups (all p's > .5).

ERP processing and analysis was completed using NetStation 4.3.1 (EGI, Inc., Eugene, OR; http://www.egi.com). First, ERP data were filtered with a 0.3–30 bandpass filter. Next, data was segmented using the segmentation tool. Segments began when the pitcher threw the ball and lasted 1000ms. Following segmentation, data were baseline corrected using 100ms prior to the pitch. Next automatic artifact detection algorithms implemented in Net Station were used to detect eye blinks, eye movements and then visually inspected. If a channel contained an eye blink or movement, it was marked as unusable for that segment. If more than 15% of channels (18 channels) were marked unusable for a particular segment, that segment was dropped from the analysis. For useable segments, each bad channel, was replaced with an average of its nearest neighbors using 'bad channel replacement' algorithms available in Net Station. Finally, all channels were re-referenced to an average of all contributing channels. Of the original 40, 32 participants had usable ERP data. ERP data was unavailable for 7 participants due to technical difficulties, (5 in the treatment group, and 2 controls), and one participant (control) due to movement artifact.

Four different kinds of events were identified and segmented separately: (1) Hits: the participant pressing the spacebar to hit the baseball when no stop signal was present, (2) Errors of omission (EOO): not pressing the spacebar on 'go' trials, (3) Error of commission (EOC): the participant erroneously pressing the spacebar following the stop signal, and (4) Correct inhibition (CI): the participant correctly withholding their response following a stop signal. EOC and CI trials were of primary interest for analysis. The mean number of EOC segments for all participants across both sessions was 29; the mean number of CI segments for all participants was 22.

Previous literature indicates that in typically developing children and adults successful stopping on an SST is associated with an N200 [32]. The N200 to successfully inhibited stop signals on an SST is of significantly lower amplitude and longer latency for children with ADHD compared to controls [30]. The N200 is an ERP component that reaches peak amplitude over frontal electrode sites roughly between 200 and 350 ms after stimulus onset, which in this study is the onset of the stop signal [69]. Each subject's ERP response on EOC and CI trials were averaged. Based on previous literature [70], channel groupings over fronto-central scalp (6, 13, 29, 35, 110, 111, 112) were selected for the N200 electrode montage, which corresponds to FCZ, FC1, FC2, FC3, FC4 within the 10–10 electrode system. For each participant, the amplitude and latency to peak between 200 and 350 ms was extracted using statistical extraction tools in NetStation. Once electrodes of interest and the time window for a particular component have been identified the statistical extraction tool automatically identifies the max amplitude and latency to max amplitude for each segment for each participant.

**Computerized training.** Gameplay for both the treatment and control groups involved a set of three SST's with varying overarching stories, training game dynamics, and characters (Fig 3). Unlike many tasks which target stopping behavior in the context of a simple go response, these tasks all targeted stopping behavior while varying the type of 'go' response across task. The changes to the task which made the game 'adaptive' or 'non-adaptive' were identical across task, which we describe for each task below. The front-end gameplay dynamics and general task parameters were consistent across both the adaptive treatment group and the control group. The control group's training experience differed from the treatment group firstly in that the latency of the stop signal for the control group did not adjust across individuals or across trials within a session—instead being at a fixed latency at the beginning of each trial and secondly that the control group had their stop signal trials on every fifth event rather

than distributing the stop-signal trials in an unpredictable manner based on their probabilities. For the adaptive treatment group, in addition to having a set probability on each event of that trial being a stop event, the stop-signal latencies for each module were determined in an individualized and adjusting manner for each patient according to proprietary methods developed by NeuroScouting LLC. These methods adjusted each day's training to focus around the participant's current IC performance level and then adjusted these latencies within a session to alter the trial-by-trial stop difficulty based on each participant's training history and current performance.

In the *Baseball game*, the pitcher would release the ball and it would approach the strike-zone, outlined by a green box, at a steady rate. The goal was to hit the ball in the strike-zone, by tapping the iPad within the green box, on each pitch that the ball remained white (go trials). The ball turning red served as the signal to not tap the iPad (stop trials). During the training phase, each session had 180 trials (or "pitches"). Each pitch lasted 750ms and there was a 1500ms intertrial interval (ITI) between each pitch. Sixty percent of the pitches were "go" trials in which players intercepted the ball with an iPad screen press when it reached the virtual homeplate. The other 40% of trials were stop-signal trials during which the latency of the stop signal (i.e. the ball turning red) varied across pitches. In the non-adaptive version used for controls and at baseline and endline assessments, the time following the pitch that the ball turned red (the stop signal) varied so that it occurred in equal bins across 10, 35, 40, 45, 50, 55, 60, 65, 75% of the duration of the time it took for the ball to travel from the pitcher to the strike zone, with an average of a stop signal occurring 48.33% of the way through the pitch. When the stop signal happened further from the ball release (e.g., at 60 or 75% of the duration), withholding a response was more difficult. When the stop signal happened earlier (e.g. 10–35% of the duration), withholding a response was easier. For the non-adaptive version of the Baseball game that were administered at baseline and endline assessments, errors of commission, errors of omission, and PSE (described below) were used to measure behavioral performance.

For the adaptive version in the treatment group training, the timing of the stop signal was related to participant performance, becoming increasingly difficult (i.e., further from ball release) as performance improved. One measure of behavioral performance on this task was PSE, which is the percent of the trial length at which delivering a stop signal results in successful inhibition on 50% of trials. Thus a PSE of 0.4, means that participant is at chance performance when the stop signal happens 40% of the way through the trajectory of the baseball from the mound to the plate. The participant's score was tracked and shown in the lower right corner.

In the *Fish game*, participants were instructed to drag a fish across the screen towards a can of fish food to feed it. Participants placed their finger on the fish avatar (named Swimmy) to activate each trial. The objective was to guide Swimmy (by keeping a finger on top of the fish as it moved across the screen) towards the fish food that was placed at equal distances but different locations within the underwater environment. As the trial starts, the fish begins swimming towards the food container and the participant is required to track the moving fish with their finger through a moving swipe on the iPad screen. On "go" trials (60% of the trials), if the participants keep their finger on top of the moving fish until it reaches the food reward, Swimmy is successfully fed. If their finger does not remain on top of the fish, Swimmy will become frustrated and swim away, thus ending the trial. On stop-signal trials (40% of the trials), Swimmy will notice something wrong with the food at some point during the swim and indicate by turning color that the participant should "release" by removing their finger from the iPad screen and allowing Swimmy to swim away from the food. Each successful feeding or non-feeding (if a stop trial) led to an increase in Swimmy's strength and higher probability of successfully leaping through a ring above the water to grab gold coins as reward. During the

training phase, there were 150 trials in each session. Each trial lasted 1250 ms and following the outcome, Swimmy would swim off the screen for a 500 ms ITI until re-appearing and the next trial began. For an average of every 45 seconds on task, there would be a break in the gameplay to allow Swimmy to jump above the water to attempt to grab a coin reward and his success in obtaining the reward was based on the accuracy of each participant in the intervening time window since the last jump. PSE was calculated by distance from the start point for the fish. For non-adaptive gameplay in the control group, the stop signal occurred an average of 55% of the way through a trial length with equal bins across 41.25, 49.5, 55, 60.5, and 68.75% of the trial length.

In the *Catapult game*, participants held their finger down on the screen to raise the power level on a gauge. When the gauge reaches the 'launch zone' (a small segment of the gauge indicating the 'go' time), the participant releases their screen press to launch a block that flies across the screen (and over a river of lava) onto a building tower of blocks. As they are pressing to raise the gauge and launch the block, if they see a stalactite fall from the top of the cavernous scene (which occurs with a simultaneous change of color of the gauge to red) above the growing tower, they do NOT release that block (holding onto the block currently in the catapult) to allow the stalactite to join the tower and build the tower faster than if it was only comprised of blocks (if they do not stop on the trial their block smashes the bigger stalactite). As the tower builds to a threshold height, it tips over and forms a temporary bridge that allows the participant's avatar to cross the lava and collect gold coins that have accrued. During the training phase, there were 284 trials per session, each one lasting 1000ms with a 1500ms ITI between each launch event. PSE was calculated by level of the gauge when the stalactite fell. For non-adaptive gameplay in the control group, the stop signal occurred an average of 55% of the way through a trial length with equal bins across 41.25, 49.5, 55, 60.5, and 68.75% of the trial length.

Before taking the games home, a practice session and individualized game-play schedule was completed. Each training session was approximately 13–15 minutes (depending on how the participant was doing, how long they spent on breaks, etc.) with a maximum of 1 session per day. The number of trials completed in each game was matched between the treatment and control conditions (Baseball: 180 trials; Catapult: 284 trials; Fish: 150 trials). A new SST game version was played each week with the same order for all participants: Baseball game, Fish game, Catapult game, and then the participant's choice for the fourth week.

### Analysis plan

For each dependent measure collected at baseline and endline (ADHD symptoms, EEG relative power in the theta bands, and ERP amplitude and latency), a repeated measures analysis of variance (ANOVA) was conducted with the within subjects factor of time (baseline vs. endline) and between subjects factor of treatment group. If any variables of interest differ between the treatment and control groups, the analyses were performed again controlling for the mismatched variables. Both results are reported when these covariates changed results. Group differences in baseline characteristics and training compliance were examined using independent-samples t-tests. Within the training group, improvement in game play was determined by first computing a difference score between first game and last game performance (based on PSE) and then subjecting this difference score to a t-test. All analyses were performed in SPSS (IBM).

### Results

### Training compliance and engagement

Number of training sessions completed differed significantly between groups ($t(38) = 3.32$, $p < .01$, $d = 1.08$). Children in the treatment group completed an average of 23.2 training

sessions, while the control group completed an average of 20.2 training sessions. Of the 20 assigned to the treatment condition, 18 participants completed a minimum of 20 training sessions, and 2 participants completed 17 sessions. Of the 20 participants assigned to the control condition, 15 completed a minimum of 20 training sessions, 4 participants completed 19 training sessions, and 1 participant completed 10 training sessions.

Using a Likert-scale rating, participants were asked to rate "How much did you like playing our games?" with 5 being "A Lot!" and 1 being "Didn't like" the mean rating was a 3.59 (SD = 1.12). There was no difference in ratings of how much the games were enjoyed between treatment (M = 3.58, SD = 1.17) and control (M = 3.60, SD = 1.10) participants.

## Task performance

To assess near transfer both groups completed a non-adaptive version of the baseball game at baseline and endline assessment sessions. Results revealed improvement in SST performance, such that errors of commission reduced from pre-training (M = 36.69, SE = 1.25) to post-training (M = 33.00, SE = 1.37), $F = 11.31$, $p < .01$. However, this effect did not differ between treatment and control groups, $p = .16$. Similarly, errors of omission reduced from pre-training (M = 14.42, SE = 1.42) to post-training (M = 10.55, SE = 1.49), $F = 5.50$, $p = .02$. This effect did not interact with treatment group, $p = .91$. There was no change in PSE from pre- to post-training nor was there an interaction between treatment and change in PSE (all $p$'s > .2). Only training related reductions in errors of commission survived Bonferroni correction (equivalent to an unadjusted $\alpha = .02$).

For the treatment group with adaptive game play, we also measured change in PSE across training. A difference in performance score for each game (i.e. baseball, fish, and catapult) was calculated by obtaining the difference in the PSE measure from the first and last sessions played during training for a given game. Differences in performance for all three games were averaged to obtain an overall performance score. Within the training group, overall performance significantly improved ($t(19) = 5.17$, $p < .01$, $d = 2.37$).

Individually, significant improvement as measured by PSE was evident for the fish game ($t(19) = 3.94$, $p < .01$, $d = 1.81$) from the first (mean PSE = 40.83) to the last (mean PSE = 61.14) training sessions for that game. Additionally, significant improvement was observed in the catapult game ($t(19) = 7.13$, $p < .01$, $d = 3.27$, First mean PSE = 41.62, Last mean PSE = 71.43). However, no difference between first (mean PSE = 46.07) and last (mean PSE = 41.28) training sessions of the baseball game ($t(19) = 1.01$, $p = .33$) were observed. Improvements in PSE on the fish and catapult games remain significant following Bonferonni correction (equivalent to an unadjusted $\alpha = .02$).

## Symptoms of ADHD

At baseline, inattention symptoms did not differ between treatment and control groups, as indicated by parental report on the SNAP-IV, parental report on the Conners, and teacher report on the Conners, ($p$'s > .6). There was a trend toward a treatment associated reduction in parent-rated average inattention symptoms on the SNAP-IV, as demonstrated by a treatment group (adaptive vs. non-adaptive) by time (baseline to endline) interaction, $F = 3.80$, $p = .06$, $\eta^2 = .09$ (Fig 4). This trend was replicated in the parental report of inattention symptoms on the Conners, $F = 3.46$, $p = .07$, $\eta^2 = .08$ (Fig 4). This trending interaction resulted from a relatively larger decrease in inattention symptoms in the treatment group (see Table 2 for symptoms).

Because the inattention subscales for the SNAP and Conners are comprised of a small number of items score (6 items from Conners; 9 items from SNAP-IV), a post-hoc analysis was

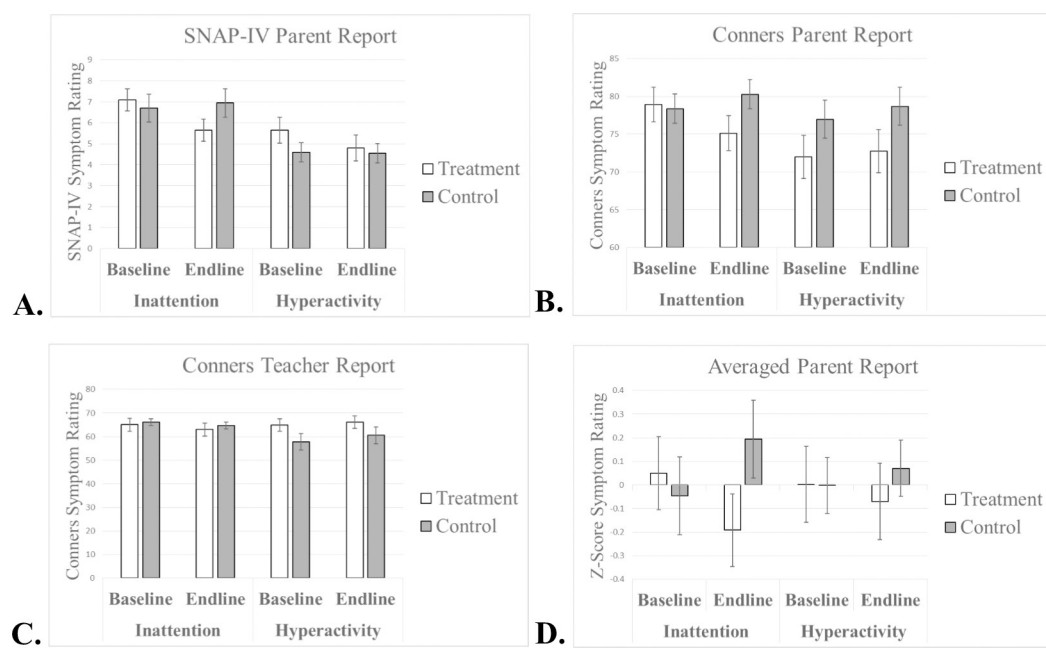

**Fig 4.** A. Parent ratings of inattention and hyperactivity as measured by the SNAP-IV. B. Parent ratings of inattention and hyperactivity as measured by the Conners. C. Teacher ratings of inattention and hyperactivity as measured by the Conners. D. Parent ratings of inattention and hyperactivity after being averaged and normed within this sample. Error bars represent within-subjects 95% confidence intervals [71].

conducted in which parent-reported scores from SNAP-IV and Conners were normed and averaged, yielding a Z-score that served as the average parental-report of inattention symptoms. Increasing the number of items contributing to a sum score serves to reduce noise, which in turn increases sensitivity to detect effects. A pearson correlation revealed a strong relationship between the inattention subscales of the Conners and SNAP-IV (Baseline: $r = .68$, $p < .01$; Endline: $r = .75$, $p < .01$), indicating that as intended, the inattention subscales of the Conners and SNAP-IV are measuring similar underlying constructs in this sample. An ANOVA conducted on the average parental-report of inattention symptoms revealed a significant treatment group (adaptive vs. non-adaptive) by time (baseline to endline) interaction, $F = 4.39$, $p = .04$, $\eta^2 = .10$ (Fig 4).

**Table 2. Treatment effects on parent-report and teacher-report of behavior.**

|  | TREATMENT GROUP | | | CONTROL GROUP | | |  |
|---|---|---|---|---|---|---|---|
|  | Pre | Post | Change (*p*) | Pre | Post | Change (*p*) | Time x Group Interaction (*p*) |
| **Teacher Conners Inattention** | 65.0 ± 12.2 | 63.0 ± 11.7 | .48 | 66.1 ± 14.0 | 64.7 ± 10.9 | .33 | .85 |
| **Teacher Conners Hyperactivity** | 64.9 ± 18.9 | 66.1 ± 17.9 | .68 | 57.8 ± 20.1 | 60.6 ± 17.4 | .45 | .72 |
| **Parent Conners Inattention** | 79.0 ± 10.4 | 75.2 ± 14.2 | .12 | 78.4 ± 10.7 | 80.3 ± 12.8 | .34 | .07 |
| **Parent Conners Hyperactivity** | 72.0 ± 18.9 | 72.8 ± 17.5 | .80 | 77.0 ± 15.1 | 78.7 ± 15.0 | .52 | .80 |
| **Parent SNAP Inattention** | 7.1 ± 2.4 | 5.7 ± 3.1 | .02 | 6.7 ± 3.4 | 7.0 ± 3.5 | .72 | .06 |
| **Parent SNAP Hyperactivity** | 5.7 ± 3.4 | 4.8 ± 3.3 | .19 | 4.6 ± 3.4 | 4.6 ± 2.5 | .92 | .32 |

The treatment group showed a significant decrease in parent-reported inattention symptoms (*p* = .04) compared to controls, which was no longer significant after controlling for age and number of sessions completed.

*Notes*: SNAP = Swanson, Nolan, and Pelham-IV [64].

## Electroencephalogram (EEG)

Independent samples t-tests confirmed that there was no treatment group differences in relative theta power for any regions of interest at baseline (all $p$'s > .1). However, a treatment group by time interaction revealed that, as compared to children in the control condition, individuals in the treatment group showed significantly greater post-training decreases in relative theta power across parietal regions of interest, as described below (Fig 5).

Treatment resulted in significant reductions in relative theta power over left parietal electrodes during the eyes closed condition ($F = 7.06$, $p = .01$, $\eta^2 = .18$) and over right parietal electrodes in the eyes open condition ($F = 6.93$, $p = .01$, $\eta^2 = .18$). This effect was not significant over right parietal electrodes in the eyes closed condition ($p = .11$) or over left parietal electrodes in the eyes open condition ($p = .11$). In the treatment group, theta power across all parietal regions was in the expected direction (i.e. decreased following treatment) for both eyes-open and eyes-closed conditions, whereas parietal theta power in the control group either stayed the same or increased non-significantly from baseline to endline (see Table 3). There was no effect of treatment on theta in frontal electrodes for either eyes open or eyes closed conditions, regardless of whether or not age and sessions completed were included as covariates

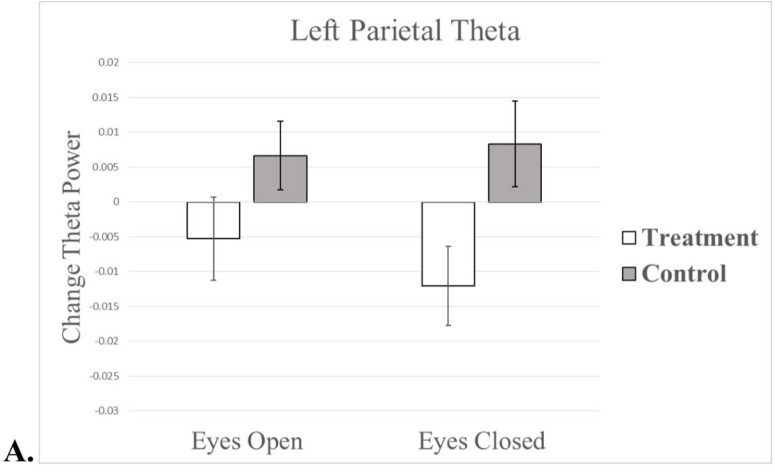

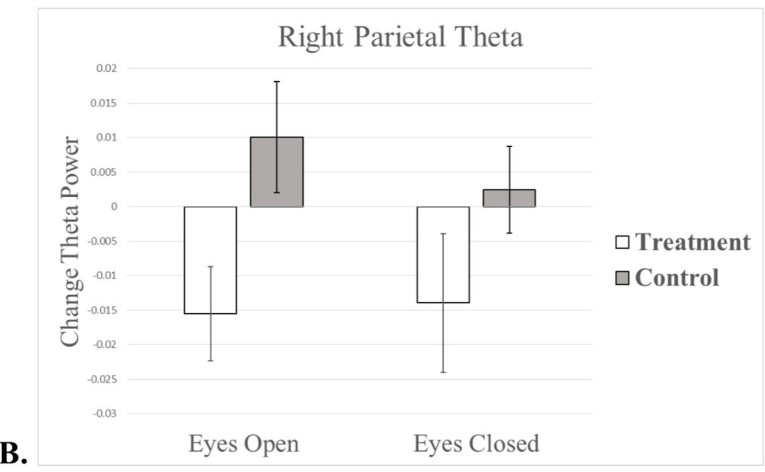

**Fig 5.** Pre-training to post-training change in eyes open and eyes closed resting state theta relative power over (A) left parietal electrodes and (B) right parietal electrodes. Error bars represent within-subjects 95% confidence intervals [71].

**Table 3. Treatment effects on resting state theta activity.**

|  | TREATMENT GROUP | | | CONTROL GROUP | | |  |
| --- | --- | --- | --- | --- | --- | --- | --- |
|  | Pre | Post | Change (*p*) | Pre | Post | Change (*p*) | Time x Group Interaction (*p*) |
| Eyes Open Left Parietal | 0.146 ± 0.038 | 0.137 ± 0.037 | < .01 | 0.137 ± 0.028 | 0.142 ± 0.039 | .22 | .11 |
| Eyes Open Right Parietal | 0.146 ± 0.040 | 0.126 ± 0.029 | < .01 | 0.130 ± 0.027 | 0.138 ± 0.052 | .19 | .01 |
| Eyes Closed Left Parietal | 0.151 ± 0.063 | 0.138 ± 0.065 | < .01 | 0.132 ± 0.052 | 0.138 ± 0.062 | .19 | .01 |
| Eyes Closed Right Parietal | 0.143 ± 0.067 | 0.129 ± 0.059 | < .01 | 0.134 ± 0.070 | 0.134 ± 0.073 | .70 | .10 |

Groups did not differ on relative theta activity at baseline (all p's > .1). Reported *p*-values for change reflects results of a paired-samples t-test. Reported *p*-values for the time by group interaction reflect results of an ANOVA as described in the analysis section.

(all *p*'s > .1). No treatment effects on theta survive Bonferroni correction (equivalent to an unadjusted α = .006).

### Event-related potential (ERP)

At baseline, there were no group differences in amplitude or latency of the N200 on any trial types (all *p*'s > .1). A treatment group by time interaction revealed an effect of adaptive training on N200 latency for CI trials ($F = 5.12$, $p = .03$, $\eta^2 = .20$). For CI trials, while there were no differences in latency for the N200 to reach peak amplitude at baseline (treatment = 240 ms; control = 256 ms), following training, the average N200 latency was longer in the treatment group (271 ms) than in the control group (237 ms). For the adaptive training group, the change from pre to post treatment in N200 latency was significant ($t(14) = 2.27$, $p = .04$, $d = 1.21$) but this change was non-significant for the non-adaptive control group ($t(16) = 1.12$, $p = .28$). As expected, there were not treatment effects on N200 latency for EOC nor were there effects on N200 amplitude for CI or EOC trials (all *p*'s >.1). The treatment effect on N200 latency does not survive Bonferroni correction (equivalent to an unadjusted α = .013).

### Relationships among treatment effects

Follow-up analyses were conducted to explore changes in ADHD symptoms, behavioral changes in IC, and putative neural mechanisms before and after treatment. Within the treatment group, greater reduction in inattention symptoms on the Conners between baseline and endline was significantly associated with increases in IC as measured by changes in number of EOC's on the non-adaptive baseball SST, $r(19) = .67$, $p < .01$. Additionally, reduction in the averaged parental inattention score was significantly associated with increases in IC as measured by changes in number of EOC's on the non-adaptive baseball SST, $r(19) = .49$, $p = .04$. Relationships between treatment-related reductions in inattention symptoms on the SNAP-IV and inhibitory control as measured EOCs were in the same direction but did not reach significance, *p*'s > .1. Treatment related changes in inattention symptoms were not significantly correlated with treatment related reductions in parietal theta or N200 latency, all *p*'s > .1. The relationship between treatment-related changes in number of EOC's and reductions in parent-reported inattention on the Conners, but not averaged parental report of inattention, survived Bonferonni correction (equivalent to an unadjusted α = 0.006).

### Covarying for age and sessions

In this study, random assignment was implemented so that systematic differences would not be expected. Although the treatment and control groups were randomized, the groups did differ on two variables: age and number of sessions completed. While there is still debate and

heterogeneity regarding inclusion of baseline characteristics as covariates in randomized control trials [72], here we elect to also report any results that changed with adjustment for age and sessions with some caveats. It has been argued that covariates should be selected a priori and also be strong predictors of the treatment outcome [73, 74]. Neither of these predictors were selected a priori or hypothesized to relate strongly to treatment outcome. To test whether treatment effects (i.e. changes in outcome variables) were related to age and sessions, exploratory analyses were conducted. Specifically, Pearson correlations were conducted to test for any potential correlations between the covariates (age and sessions) and changes in the outcomes of interest (theta reductions, inattention reductions, changes in n200 latency, and increases in behavioral inhibitory control). For all outcome variables, except theta power over right parietal electrodes, the change in outcome variable did not correlate with age, making it more likely that age is not driving treatment effects. In the one case where age was correlated with our change in outcome—greater reduction in theta oscillatory activity over right parietal electrodes during the eyes open condition ($r = .36$, $p = .04$), the significant treatment effect on right parietal eyes open theta survives controlling for age ($p < .05$), again making it unlikely that age is a confounder in this analysis. Similarly, there were not significant associations between number of sessions and changes in most outcome variables. One exception is with parent-reported SNAP-IV inattentive symptoms. Number of sessions completed and reductions in inattention symptoms as reported on the SNAP-IV were correlated ($r = .49$, $p < .01$). This was only true for the SNAP-IV and was not true for the Conners.

Overall, treatment effects were largely unchanged between adjusted and unadjusted analyses of the data with two exceptions. The interaction between treatment group and time on parental report of inattention, across the Conners, SNAP-IV, and averaged parental inattention, was no longer significant after the inclusion of these covariates ($p$'s > .1). Additionally, the interaction suggesting treatment related reductions in theta power over left parietal electrodes during eyes open condition did reach significance when age and sessions were included as covariates ($F = 6.28$, $p = .02$, $\eta^2 = .17$).

## Discussion

The present study implemented a randomized, single blind, active control design, where parents and participants were blind to their treatment condition. In this study we investigated the effects of a 4-week computerized IC training program on ADHD symptoms and neural activity in children with ADHD. It was hypothesized that training would result in reduced symptoms of ADHD and changes in neural function linked to ADHD symptoms, and these hypotheses were partially supported. A trend toward training induced reductions in parent-reported ADHD symptoms was evidenced in parent-report on the SNAP-IV and Conners. These effects reached significance following a post-hoc aggregation of the parent-reported inattention data. Additionally, training resulted in changes in neural activity, specifically reductions in resting-state theta oscillatory activity over parietal electrodes and longer latencies in N200 amplitude for correctly inhibited Stop trials. These results, while preliminary, suggest that computerized training of IC systems warrants further investigation as they may have the potential to be used as an adjunct intervention for children with ADHD.

We observed a trend where, relative to children who played non-adaptive IC games, children who played our adaptive IC training games were rated as having improved attention by their parents. This result was trending on the inattention subscales in both the Conners and the SNAP-IV. We did not observe any impact of treatment on reported symptoms of hyperactivity. While these results are suggestive, particularly given their marginal nature and that they were not large enough to withstand controlling for age and sessions, the trending pattern was

replicated across both parent-report questionnaires administered. Additionally, when inattention symptoms were examined as an aggregate parent-report score, a significant treatment effect on inattention symptoms was observed. The suggestion that IC training could impact symptoms of inattention, specifically, is consistent with findings that appropriate response selection on inhibitory control tasks requires engagement of high attentional resources to continuously monitor upcoming stimuli [75], particularly in children [76]. Indeed, a meta-analysis of neuroimaging of inhibitory control tasks has demonstrated a high degree of overlap in neural activity linked with IC and with increasing attentional demands [77]. It has further been confirmed using a latent-variable analysis that the constructs of IC on a SST (ability to stop prepotent responses) and attentional control (the ability to resist interference from distractors) are closely linked [78].

Furthermore, within the treatment group, individuals who showed the most behavioral improvement in IC, as measured by errors of commission, also showed the greatest reductions in parent-reported inattention symptoms on the Conners. Our findings are an initial step indicating that IC training alone, without working memory, has potential to reduce parental report of inattention symptoms. However, the findings from this initial, small-scale study should be replicated and extended in larger samples to address limitations described below. The lack of an effect of treatment on hyperactivity is consistent with a recent meta-analysis which found that other computerized trainings targeting neuropsychological deficits in ADHD impact parental report of inattention symptoms but not hyperactivity [54].

Relative to a non-adaptive control group, children in the adaptive training group showed reductions in both eyes-open and eyes-closed resting-state relative parietal theta power. Previous research has shown that higher theta power is linked with greater inattention, failures in inhibition, and executive problems in children with ADHD [26, 79–81]. Relatedly, medication-induced reductions in theta power are associated with reductions in symptoms of ADHD as well as improvement in cognitive functioning [82]. In typically developing participants, it has previously been suggested that power in the theta frequency band underlies IC [83], and neurostimulation over the right inferior frontal gyrus, an area of the brain strongly associated with IC [84], reduces theta oscillatory power and improves response inhibition [85, 86]. Our findings align with previous work linking IC and theta oscillatory power by demonstrating that adaptive training of IC in children with ADHD reduces resting theta oscillatory power. It is surprising that our results show training-related changes specific to parietal, but not frontal, regions. Previous work has shown aberrant activity in fronto-parietal and fronto-striatal networks during inhibitory tasks in individuals with ADHD [18, 87]. Regions across the fronto-parietal network are involved in inhibitory control, and particular emphasis has been placed on the frontal regions involved in this task, which makes the lack of an effect in frontal regions unexpected [88]. Furthermore, theta power across the fronto-parietal network is thought to underlie aspects of cognitive control [89], and simultaneous EEG-fMRI recording has linked theta oscillatory activity with widespread activity across frontal and parietal regions [90]. Given this prior work implicating the fronto-parietal control network in IC, it was hypothesized that effects would be observed in both frontal and parietal regions. Further work is needed to investigate the regional specificity of training IC related changes in theta-oscillatory activity.

Adaptive IC training also impacted another neural measure related to IC, the N200 component. The N200 is a frontocentral component which is associated with inhibition on an SST [32]. This component has been suggested to play a role in attentional allocation and inhibition of motor processes [91]. During successfully inhibited trials, children with ADHD show reduced amplitude and longer latencies of the N200 compared to controls [30]. We observe that adaptive IC training increased N200 latency for successfully inhibited trials on an SST in

children with ADHD. This result may seem counterintuitive, as it is not a 'normalization' of the commonly observed difference between children with and without ADHD. However, the adaptive training group (as compared to control) experienced increasingly longer latencies between the go cue and stop-signal as performance improved. Thus, if the N200 is tracking successful inhibition, it is possible that the actual time prior to successful inhibition increased for children in the adaptive group as a function of training. It is also possible that this increase in latency for the N200 simply reflects some kind of compensatory neural mechanism which is evident already when comparing children with and without ADHD and which is further enhanced by adaptive IC training. Future work should attempt to disambiguate the meaning of theses observed changes in N200 latency.

While we observed change in potential neural mechanisms, theta power and N200 latency, and trending differences in symptoms of inattention, we observed few examples of adaptive-training-specific near transfer or change in task performance. Overall task performance improved pre to post treatment but the degree of improvement did not differ by treatment group. Interpretation of this finding should be moderated by our observations of task performance during the training for the treatment group. Behavioral improvements on the SST's were not shown by the adaptive training group for the Baseball game, but were shown in the latter two games (Catapult and Fish). We chose the Baseball game for our test of near transfer across both groups and it may be that variation in task demands across games lead to improvements in this game being more difficult to observe.

While our results suggest that IC training in ADHD warrants further investigation as a potential adjunct treatment, there are also limitations to the current study. Firstly, we do not observe impact of training on teacher report of inattention. The lack of findings could be because parent report was more biased than teacher report; however, this is less likely in our sample given that both the participant and parent were blind to study condition, and our results, while trending, are specific to the same subscale on two different inventories of ADHD symptoms. For example, no significant reduction in symptoms was shown for participants in the non-treatment group, and no reduction in hyperactivity across either group was reported. An alternative explanation could be that teachers may not see improvements because they only see the children at school when the children are medicated, whereas parents are more likely to observe their children off medication. Consistent with this possibility, teachers overall report fewer symptoms than parents in this sample (Table 2). In addition, some studies find that teachers report fewer inattention symptoms than parents overall and parent-teacher inter-rater reliability is worse for inattention symptoms than for hyperactivity symptoms [92]. Thus teachers may not be as reliable reporters as parents in this case.

Secondly, our results indicating training-related changes in parent-reported inattention symptoms are promising but not definitive. These changes, examined separately in the Conners and SNAP-IV, are trending but do not reach statistical significance unless examined in aggregate form. Given that this trending effect is replicated in two different surveys and it reaches significance when examined in aggregate, an investigation of the effects of IC on inattention symptoms in a larger sample of children with ADHD is warranted. Although the randomly assigned groups did not differ on a number of important attributes, the two groups did differ in average age and number of training sessions completed. Interestingly, the impact of training on metrics of neural function (i.e. resting-state theta activity and N200 latency) during inhibition were robust to controls for age and training sessions. However, treatment effects on parental report of inattention symptoms did not remain significant when controlling for age and sessions. Further, given the small sample size and diagnostic makeup of the sample, this study cannot differentiate whether the observed effects may differ by subtype. Our sample consisted of primarily participants who met criteria for ADHD Inattentive or Combined subtypes,

consistent with reports that these subtypes are most common in children in this age group [63]; however, this limits our ability to determine whether the current findings would generalize to individuals with a primary diagnosis of ADHD Hyperactive subtype.

Finally, future work is required to better elucidate the neural mechanisms through which IC training leads to improvements in inattention. Our findings suggest that IC training reduces parietal theta oscillatory power and shifts the neural correlates of successful inhibition. While a reduction in theta oscillatory power and neural function during IC are two plausible mechanisms through which IC training might impact inattention, these findings are preliminary, completed in a small sample, and require replication. Additionally, this study specifically investigated IC training effects in children with ADHD, because of hypothesized IC deficits in children with ADHD [11, 12]; however, whether IC training in typically developing children would also lead to improvements is unclear. There is some indication that training programs show larger effects in individuals with cognitive deficits than individuals without deficits [93, 94, but see also 95], and an investigation of IC training in typically developing preschoolers did not show far transfer effects [96], suggesting these results may be strongest in individuals with ADHD. Future work should investigate the neural mechanisms of changes in inattention and how sustainable these changes are over time.

As a randomized control trial testing computerized training selectively targeting IC in children with ADHD, this study provides initial support for the possibility of using computerized IC training in conjunction with other ADHD treatments to help alter behavior and the underlying neural mechanisms of ADHD. Consistent with theories that IC deficits are central to ADHD [11, 12], adaptive IC training led to changes in neural markers linked with IC and consistent trends toward ADHD inattentive symptom reduction after just four weeks. In light of the carefully matched controls, who also played games requiring inhibitory control, and small sample sizes, these initial results are particularly impressive. Furthermore, the computerized game format was well liked by participants and has the convenience of being played in the home, which likely contributed to the high adherence to training schedules. In summary, these initial results are promising and indicate that further work should investigate IC training in ADHD.

## Supporting information

**S1 Checklist. CONSORT 2010 checklist of information to include when reporting a randomised trial**\*.
(DOC)

**S1 Dataset.**
(XLSX)

**S2 Dataset.**
(XLSX)

**S1 Flow diagram. CONSORT 2010 flow diagram.**
(DOC)

**S1 Protocol.**
(DOC)

## Acknowledgments

We thank our child and family participants. Poster version of training effects on relative theta power and parent ratings of inattention presented at the Second Annual Flux Congress.

## Author Contributions

**Conceptualization:** Brian Miller, Wes Clapp, Margaret A. Sheridan.

**Data curation:** Marcus Way, Katrina Bridgman-Goines, Margaret A. Sheridan.

**Formal analysis:** Kristin N. Meyer, Rosario Santillana, Marcus Way, Margaret A. Sheridan.

**Funding acquisition:** Margaret A. Sheridan.

**Investigation:** Rosario Santillana, Marcus Way, Katrina Bridgman-Goines, Margaret A. Sheridan.

**Methodology:** Marcus Way, Margaret A. Sheridan.

**Project administration:** Rosario Santillana, Margaret A. Sheridan.

**Resources:** Margaret A. Sheridan.

**Software:** Brian Miller, Wes Clapp.

**Supervision:** Margaret A. Sheridan.

**Validation:** Margaret A. Sheridan.

**Visualization:** Marcus Way, Margaret A. Sheridan.

**Writing – original draft:** Rosario Santillana, Margaret A. Sheridan.

**Writing – review & editing:** Kristin N. Meyer, Brian Miller, Wes Clapp, Margaret A. Sheridan.

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
