## [Decision Letter · Decision Letter 0]

27 Aug 2019

PONE-D-19-18201

Computer-based inhibitory control training in children with attention-deficit/hyperactivity disorder (ADHD): Evidence for behavioral and neural impact.

PLOS ONE

Dear Dr Sheridan,

Thank you for submitting your manuscript to PLOS ONE. After careful consideration, we feel that it has merit but does not fully meet PLOS ONE’s publication criteria as it currently stands. Therefore, we invite you to submit a revised version of the manuscript that addresses the points raised during the review process.

The manuscript has been assessed by two reviewers. The reviewers have raised concerns that need attention in a revision, they request clarifications on a number of aspects of the design and reporting. Please carefully revise the manuscript to address the comments raised by the reviewers.

In addition to the items raised by the reviewers, could you please further discuss the power of the study, the study protocol listed a sample size of 25 participants per group, but the manuscript reports 20 in each arm of the trial.

We would appreciate receiving your revised manuscript by Oct 08 2019 11:59PM. Please include the following items when submitting your revised manuscript:

We look forward to receiving your revised manuscript.

Kind regards,

Iratxe Puebla

Senior Managing Editor, PLOS ONE

**Journal Requirements:**

"I have read the journal's policy and the authors of this manusciprt have the following competing interests: Authors Brian Miller and Wes Clapp own Neuroscouting, LLC. All other authors report no conflicts of interest."

3. Thank you for submitting your clinical trial to PLOS ONE and for providing the name of the registry and the registration number. The information in the registry entry suggests that your trial was registered after patient recruitment began. PLOS ONE strongly encourages authors to register all trials before recruiting the first participant in a study.

1) your reasons for your delay in registering this study (after enrolment of participants started);

2) confirmation that all related trials are registered by stating: “The authors confirm that all ongoing and related trials for this drug/intervention are registered”.

Please also ensure you report the date at which the ethics committee approved the study as well as the complete date range for patient recruitment and follow-up in the Methods section of your manuscript.

**Comments to the Author**

1. Is the manuscript technically sound, and do the data support the conclusions?

Reviewer #1: Partly

Reviewer #2: Yes

2. Has the statistical analysis been performed appropriately and rigorously? 

Reviewer #1: Yes

Reviewer #2: Yes

3. Have the authors made all data underlying the findings in their manuscript fully available?

Reviewer #1: Yes

Reviewer #2: Yes

4. Is the manuscript presented in an intelligible fashion and written in standard English?

Reviewer #1: Yes

Reviewer #2: Yes

5. Review Comments to the Author

Reviewer #1: The study presents a pre-post investigation of cognitive training on ADHD participants. The study has relevant contributions to present, but some issues may need to be addressed. Ultimately, I am not sure that the number of participants is enough to establish reliable results independent of the confounds of age and number of sessions. Though the authors show that outcomes did not correlate with these variables, the change/difference from pre and post variables, did. The following issues and questions are intended to help clarify these questions, and other minor issues.

1. Researchers were not blind to Treatment x Control groups. Considering that the results are based on parents and teachers reports, who were blind to the groups, I am not sure the single-blind procedure would be an issue; however, the Conner’s Teacher Rating Scale is administered by a third party, was this third party blind to the groups? If not, please clarify and describe how the person administering the Scale would avoid being influenced by the knowledge of the group and possibly influencing the response from teachers and parents.

2. Were there any differences in SNAP or other pre-training evaluations among groups? The authors report there were age differences. Please address all measures and report whether any were significantly different in pre-training.

3. The authors mention this would be the first CT study on inhibitory control, but there is a recent study on CT and IC (among other executive functions), and brain imaging, that should be cited. (1)

1. de Oliveira Rosa V, Rosa Franco A, Abrahão Salum Júnior G, Moreira-Maia CR, Wagner F, Simioni A, et al. Effects of computerized cognitive training as add-on treatment to stimulants in ADHD: a pilot fMRI study. Brain Imaging Behav [Internet]. 2019 Jun 19 [cited 2019 Aug 5];1–12. Available from: http://link.springer.com/10.1007/s11682-019-00137-0

4. For clarification: thought the treatment and control training sessions had the same number of trials (e.g. 180 for the baseball game), the control group completed fewer trials (as reported in results). Why did controls perform fewer trials? They enjoyed the games just as well, so any explanations (or even speculation)?

5. The study does not have non-ADHD control and treatment groups; evidently, it is not so that the authors should collect more data, but maybe this shortcoming should be addressed together with recent literature on CT training with non-ADHD participants; what are the neural effects on non-ADHD? How do they correlate to performance improvements and generalization to other tasks (if any)?

6. The p-value for the significant decrease in parent-reported inattention is barely significant. Please address this result in relation to question 1 above.

7. Both groups showed improvement in errors of commission and omission; so, what is the role of the adaptability, or of training inhibitory control? It seems the effects is more associated with age and sessions, as the authors report that these variables correlated with the changes in outcomes. The authors state that these variables did not correlate with the outcome, but isn’t the change in inattention, latency, IC, etc what matters? Please address and clarify, as it seems that age and sessions do play a role in the change from pre and post, so I am not sure that the absence of correlation with the outcomes (which is an instance, a cross-sectional result, rather than the representation of change from pre and post) eliminates the possibility that sessions were playing a role. Please clarify why an increased number of trials/sessions would not influence treatment effects? Of course, there may be an optimal number of sessions for any CT, but is this the case?

8. The authors should better foreshadow the literature in CT and executive functions or include more of it in the Discussion; I missed, at least, a discussion of the role of training with an adaptable game as was the case and the actual cognitive mechanism involved? It seems there is a wealth of literature on CT that could be included, such as the Jaeggi and colleagues and more recent work from Rubia and colleagues.

Reviewer #2: In this randomized controlled trial in children with ADHD, the authors explored a potential adjunct to the conventional treatment options (medication, cognitive behavioral therapy). Inhibitory control (IC) was targeted using a modified stop-signal task training. Children were randomly assigned to adaptive treatment or non-adaptive control with identical stimuli and task goals, and trained at home for four weeks. Effects on ADHD symptoms and neural activity were measured and compared between the two groups.

Although I cannot comment on the content, I really enjoyed reading this interesting paper on an important topic—finding ways to supplement conventional therapies for ADHD which may be well received by children to whom they are offered. The analyses as described are appropriate and well explained, although I have a few comments on the tables (below). The authors have done a good job of remaining cautious about their findings, and the conclusions they offer do not extend beyond the limitations of the study.

Minor comments:

It would be helpful to define “near” and “far” transfer improvement for readers who are not familiar with these concepts.

Also, “N200” is not really defined until the discussion section. It would be also be helpful to have some explanation of this early in the paper.

This sentence didn’t really make sense to me: “The N200 is a negative going component maximal over frontal scalp cites and peaks roughly between 200 and 350 ms post stimulus onset.”

Table 1. Please include p-values, since they are discussed in the text.

Tables 2 and 3. Please give within-group and between-group p-values.

6. PLOS authors have the option to publish the peer review history of their article (what does this mean?). If published, this will include your full peer review and any attached files.

Reviewer #1: No

Reviewer #2: No

---

## [Author Response · Author response to Decision Letter 0]

19 Sep 2019

Response to Reviewers: for manuscript #PONE-D-19-18201, “Computer-based inhibitory control training in children with attention-deficit/hyperactivity disorder (ADHD): Evidence for behavioral and neural impact” by Meyer, Santillana, Miller, Clapp, Way, Bridgmann-Goines, & Sheridan. 

Dear Dr. Puebla,

Please find attached a revised version of this manuscript incorporating the feedback and suggestions from reviewers. Below are our detailed responses to the points raised by each of the 2 reviewers. The reviewers’ comments are shown in italics, and our replies are directly below each comment with reference to the sections edited. 

In addition to the specific comments by the reviewers, documented below, we read carefully the editors comments, 

“In addition to the items raised by the reviewers, could you please further discuss the power of the study, the study protocol listed a sample size of 25 participants per group, but the manuscript reports 20 in each arm of the trial.”

Our initial plan was to collect 25 participants per arm of the study; however, due to time and monetary constraints, only 20 participants were randomized to each arm of the trial. It should be noted, our power analysis reported in the study was based on implementing this training with healthy individuals. As such this may have been a modest estimate of effect size given that the current study tests effects of inhibitory control training in individuals with a specific deficit in inhibitory control. For example, based on the large effect sizes in a study of working memory and inhibitory control training in individuals with ADHD (1), a sample of just 10 individuals with ADHD in each arm would give 80% power to detect effects on parent-reported inattention symptoms and resting-state theta oscillatory power. To further address concerns about power, we have ensured that effect sizes are reported for significant results.

Reviewer #1:

1. The study presents a pre-post investigation of cognitive training on ADHD participants. The study has relevant contributions to present, but some issues may need to be addressed. Ultimately, I am not sure that the number of participants is enough to establish reliable results independent of the confounds of age and number of sessions. Though the authors show that outcomes did not correlate with these variables, the change/difference from pre and post variables, did. The following issues and questions are intended to help clarify these questions, and other minor issues.

REPLY: We appreciate this reviewer pointing out the importance of testing correlations between the change in variables with the potential confounds of age and sessions. This comment helped us to recognize that our language may not have been clear in the “Covarying for Age and Sessions” section that the relationship between change in outcome variables with age and sessions completed was indeed what we were testing, and we apologize for the confusion. We have clarified our language in that section to be more explicit (pages 26-27). 

As should now be more clear in that section, we largely do not see a relationship between change in outcome variables with either age or sessions completed. There are two exceptions to this general rule. The first is that age is related to change in theta oscillatory activity over right parietal electrodes during the eyes open condition; however, the significant treatment effect on right parietal eyes open theta survives controlling for age (p < .05), making it unlikely that age is driving these effects. The second exception is that change in parent-reported inattentive symptoms on the SNAP-IV is correlated with number of sessions completed. This relationship is not present for parent-reported inattentive symptoms on the Conners. A priori, we would not expect a session effect confound to be present for just one measure of inattention, particularly given the strong relationship between the SNAP-IV and Conners. However, we additionally examined the relationship between sessions and change in parent-reported inattentive symptoms on the SNAP-IV by group. This analysis demonstrated that the relationship between sessions and parent-reported inattentive symptoms on the SNAP-IV was actually only present in the control group (r = -.71, p < .01) but not the treatment group (p = .72). Therefore the significant change in inattentive symptoms on the SNAP-IV evidenced in the treatment group, t(19) = 2.68, p =.02, is unlikely to be driven by number of sessions. Further, the change in inattentive symptoms on the SNAP-IV was not present in the control group (p = .72). As such, it is unlikely that the difference in sessions between groups is driving a difference in changes in inattentive symptoms reported on the SNAP-IV. We report our findings with and without these covariates for the sake of transparency, but also highlight the caveat that there is still debate about inclusion of covariates stemming from group differences in randomized control trials (2), particularly when the covariates are neither selected a priori nor strong predictors of treatment outcome (3,4).

2. Researchers were not blind to Treatment x Control groups. Considering that the results are based on parents and teachers reports, who were blind to the groups, I am not sure the single-blind procedure would be an issue; however, the Conner’s Teacher Rating Scale is administered by a third party, was this third party blind to the groups? If not, please clarify and describe how the person administering the Scale would avoid being influenced by the knowledge of the group and possibly influencing the response from teachers and parents.

REPLY: In an effort to reduce the possibility for bias being introduced into the study, we minimized interaction with individuals completing the Conner’s Teacher Rating Scale. The Conner’s Teacher Rating Scale is a questionnaire with printed instructions on the document. There was no verbal interaction with individuals completing this questionnaire beyond asking them to complete this questionnaire and subsequently the form was emailed to them for them to complete.

3. Were there any differences in SNAP or other pre-training evaluations among groups? The authors report there were age differences. Please address all measures and report whether any were significantly different in pre-training.

REPLY: There were no differences in SNAP, Conners, or any other measure at baseline. The one exception was age which did differ between treatment and control groups at baseline. This is why we include analyses with and without controls for age. We apologize if this was confusing: our analysis of differences at baseline in demographic characteristics is reported in the “Participants and Procedure” section of the Methods. However, all other analysis of potential differences at baseline for outcome variables are reported in the results section for that analysis (e.g. analysis of baseline differences for ADHD symptoms is reported in the ADHD symptoms section of the results; analysis of baseline differences for EEG power is reported in the EEG section of the results, etc.). 

4. The authors mention this would be the first CT study on inhibitory control, but there is a recent study on CT and IC (among other executive functions), and brain imaging, that should be cited. 

de Oliveira Rosa V, Rosa Franco A, Abrahão Salum Júnior G, Moreira-Maia CR, Wagner F, Simioni A, et al. Effects of computerized cognitive training as add-on treatment to stimulants in ADHD: a pilot fMRI study. Brain Imaging Behav [Internet]. 2019 Jun 19 [cited 2019 Aug 5];1–12. Available from: http://link.springer.com/10.1007/s11682-019-00137-0

REPLY: We appreciate the reviewer bringing this study to our attention, and we have included discussion of this recent study in our introduction section discussing other interventions that have implemented IC training along with other executive functioning training (page 5). We have also clarified in the manuscript that we are the first study testing specific effects of inhibitory control training, rather than training aimed at an array of executive functions. 

5. For clarification: though the treatment and control training sessions had the same number of trials (e.g. 180 for the baseball game), the control group completed fewer trials (as reported in results). Why did controls perform fewer trials? They enjoyed the games just as well, so any explanations (or even speculation)?

REPLY: We interpret this reviewer’s comment that “the control group completed fewer trials (as reported in the results)” to be about our finding that the control group completed fewer number of training sessions. This is why we performed our analyses with and without controls for number of sessions. As the reviewer notes above, the training sessions had the same number of trials between groups. We find it intriguing that the control group completed fewer training sessions, however, we can only speculate as to the source of this difference. It is possible that while the degree to which control and treatment participants report liking the games is similar, that how much their enjoyment of the game changes with practice may vary across groups. Specifically, the control group experience a similar level of difficulty across sessions, whereas the treatment group experiences increase in difficulty as their performance improves across session. As such, it is possible that the control group report liking the games just as well as the control group but felt they were achieving “mastery” of the current level, which would not have occurred in the treatment group. Again, these were not explicitly tested, but may have contributed to the differences. 

6. The study does not have non-ADHD control and treatment groups; evidently, it is not so that the authors should collect more data, but maybe this shortcoming should be addressed together with recent literature on CT training with non-ADHD participants; what are the neural effects on non-ADHD? How do they correlate to performance improvements and generalization to other tasks (if any)?

REPLY: Our primary focus in this study was to provide initial data addressing the utility of inhibitory control training in a group of individuals with a selective deficit in inhibitory control, children with ADHD. As a result, including non-ADHD controls is outside the scope of the current investigation. However, it is possible that the results would differ if we had completed this training in children without ADHD, and we hope future researchers will investigate this possibility. As such, we have included further discussion of training effects in non-clinical populations in our discussion (page 33). Relatedly, comparing effects of training between two ADHD groups is common among the CT literature (e.g. Johnstone, et al., 2010; Oliveira, et al., 2019 mentioned above). 

7. The p-value for the significant decrease in parent-reported inattention is barely significant. Please address this result in relation to question 1 above.

REPLY: We report this effect because it reached significance at p < .05, as is standard convention. We appreciate the reviewer’s concern about potential relationships to age and sessions as reported in question 1. Please see the response to question 1 above in which we address age and sessions as covariates. Additionally, we now include effect sizes for all significant results to address questions as to the magnitude of effects. 

8. Both groups showed improvement in errors of commission and omission; so, what is the role of the adaptability, or of training inhibitory control? It seems the effects is more associated with age and sessions, as the authors report that these variables correlated with the changes in outcomes. The authors state that these variables did not correlate with the outcome, but isn’t the change in inattention, latency, IC, etc what matters? Please address and clarify, as it seems that age and sessions do play a role in the change from pre and post, so I am not sure that the absence of correlation with the outcomes (which is an instance, a cross-sectional result, rather than the representation of change from pre and post) eliminates the possibility that sessions were playing a role. Please clarify why an increased number of trials/sessions would not influence treatment effects? Of course, there may be an optimal number of sessions for any CT, but is this the case? 

REPLY: We agree with the reviewer that change in outcomes is most important, and again apologize for any lack of clarity in conveying our test of relationships for change in outcomes with age and sessions. To clarify, change in performance (i.e. change in errors of commission, errors of omission, and P.S.E.) did not correlate with age and sessions. We do note the reviewer’s comment that both groups showed improvement in performance, demonstrating some level of improvement in inhibitory control on the task for both adaptive and non-adaptive training. We do see differences in neural changes associated with inhibitory control between the adaptive and non-adaptive training (i.e. changes in N200 latency, changes in resting-state theta) that are not related to age or sessions. While one possibility is that the neural measures are more sensitive to a difference in change, we also consider the particular task used to measure behavior. As noted in the manuscript, the baseball game (which was used to assess behavior at baseline and endline) was played the first week for all participants, and is the only of the games for which participants in the treatment group did not show significant behavioral improvements. This relates to the reviewer’s latter point about optimal number of sessions. We do see significant improvements in behavior during training sessions for the games administered in the later weeks. Our study is not powered to determine what an optimal number of training sessions might be, but it appears possible that we see more behavioral improvement for games completed after the first week. 

9. The authors should better foreshadow the literature in CT and executive functions or include more of it in the Discussion; I missed, at least, a discussion of the role of training with an adaptable game as was the case and the actual cognitive mechanism involved? It seems there is a wealth of literature on CT that could be included, such as the Jaeggi and colleagues and more recent work from Rubia and colleagues.

REPLY: We appreciate this reviewer’s suggestion, and we have now included the suggested literature along with other work that highlights differences between adaptive and non-adaptive training (pages 4 and 5).

Reviewer #2:

1. In this randomized controlled trial in children with ADHD, the authors explored a potential adjunct to the conventional treatment options (medication, cognitive behavioral therapy). Inhibitory control (IC) was targeted using a modified stop-signal task training. Children were randomly assigned to adaptive treatment or non-adaptive control with identical stimuli and task goals, and trained at home for four weeks. Effects on ADHD symptoms and neural activity were measured and compared between the two groups. Although I cannot comment on the content, I really enjoyed reading this interesting paper on an important topic—finding ways to supplement conventional therapies for ADHD which may be well received by children to whom they are offered. The analyses as described are appropriate and well explained, although I have a few comments on the tables (below). The authors have done a good job of remaining cautious about their findings, and the conclusions they offer do not extend beyond the limitations of the study.

REPLY: We appreciate the comments and have incorporated the suggestions as detailed below.

2. It would be helpful to define “near” and “far” transfer improvement for readers who are not familiar with these concepts.

REPLY: Thank you for the suggestion, we have added clarifying definitions of these concepts in the manuscript (page 4).

3. Also, “N200” is not really defined until the discussion section. It would be also be helpful to have some explanation of this early in the paper.

REPLY: We have included a definition of the N200 and relation to inhibitory control at its first mention in the introduction (page 4).

4. This sentence didn’t really make sense to me: “The N200 is a negative going component maximal over frontal scalp cites and peaks roughly between 200 and 350 ms post stimulus onset.”

REPLY: We have clarified that sentence as follows: “The N200 is an ERP component that reaches peak amplitude over frontal electrode sites roughly between 200 and 350 ms after stimulus onset, which in this study is the onset of the stop signal.”

5. Table 1. Please include p-values, since they are discussed in the text.

REPLY: Thank you for bringing this to our attention. We have included the p-values with Table 1.

6. Tables 2 and 3. Please give within-group and between-group p-values.

REPLY: The p-values have been added for Tables 2 and 3. 

Additional Journal Requirements:

1. We have checked to ensure our manuscript meets PLOS ONE’s style requirements, including those for file naming. 

2. We have updated the Competing Interests section in the Cover Letter to confirm that we can adhere to PLOS ONE policies on sharing data and materials. 

3. We have highlighted the section detailing reasons for registering the clinical trial after enrollment had started (page 8) along with confirmation that all related trials are registered. Details of dates of approval and participant recruitment have been added (page 8).

---

## [Decision Letter · Decision Letter 1]

30 Jul 2020

PONE-D-19-18201R1

Computer-based inhibitory control training in children with attention-deficit/hyperactivity disorder (ADHD): Evidence for behavioral and neural impact

PLOS ONE

Dear Dr. Sheridan,

Thank you for submitting your manuscript to PLOS ONE. After careful consideration, we feel that it has merit but does not fully meet PLOS ONE’s publication criteria as it currently stands. Therefore, we invite you to submit a revised version of the manuscript that addresses the points raised during the review process.

I consider several of the Reviewer3's suggestions to be important for improving the article. In particular: a) include in the Introduction the existing controversy regarding the absence versus presence of results when cognitive training is applied (maybe differences in treatment efficacy is related to training time, number of sessions or other variables that have not been made explicit in the document) and b) be very careful when making statements like "the first study of its kind to use computerized training of IC in children with ADHD"; Reviewer3 indicates that this is not the case, please carefully review the literature.

We look forward to receiving your revised manuscript.

Kind regards,

Thalia Fernandez, Ph.D.

Academic Editor

PLOS ONE

Reviewers' comments:

Reviewer's Responses to Questions

**Comments to the Author**

1. If the authors have adequately addressed your comments raised in a previous round of review and you feel that this manuscript is now acceptable for publication, you may indicate that here to bypass the “Comments to the Author” section, enter your conflict of interest statement in the “Confidential to Editor” section, and submit your "Accept" recommendation.

Reviewer #1: All comments have been addressed

Reviewer #2: All comments have been addressed

Reviewer #3: All comments have been addressed

2. Is the manuscript technically sound, and do the data support the conclusions?

Reviewer #1: Partly

Reviewer #2: (No Response)

Reviewer #3: Yes

3. Has the statistical analysis been performed appropriately and rigorously? 

Reviewer #1: Yes

Reviewer #2: (No Response)

Reviewer #3: Yes

4. Have the authors made all data underlying the findings in their manuscript fully available?

Reviewer #1: Yes

Reviewer #2: (No Response)

Reviewer #3: Yes

5. Is the manuscript presented in an intelligible fashion and written in standard English?

Reviewer #1: Yes

Reviewer #2: (No Response)

Reviewer #3: Yes

6. Review Comments to the Author

Reviewer #1: No additional comments. The authors have replied to the issues to my satisfaction. I suggest that the paper be accepted.

Reviewer #2: (No Response)

Reviewer #3: This study is a pre-post investigation of inhibitory control cognitive training for children with ADHD. Such work is very important, especially now as companies are emerging, making strong claims about cognitive training and charging families of children with ADHD huge amounts of money for their cognitive training services.

The authors might consider referencing the following related work and toning down the language related to the novelty of the present study and should consider referencing Shavlev, Tsal, & Mevorach (2002; they may have more recent work as well) and Jones, Katz, Buschkuehl, Jaeggi & Shah (2018). Specifically, in the abstract, the authors say that theirs is the only study in which cognitive training for ADHD has targeted inhibition. This is not the case.

In general, the authors do a good job describing their study and their analyses. However, most of the results are null. This is not a bad thing -- we should certainly publish such findings. However, the authors should be more upfront about this in the abstract, so as not to mislead the reader. Relatedly, the introduction section would do well to discuss that a large portion of cognitive training research has produced null or small effects. There are several metaanalyses showing both that cognitive training works and that it does not. I recommend that the authors acknowledge this ongoing debate and position their work as contributing to this ongoing discussion.

7. PLOS authors have the option to publish the peer review history of their article (what does this mean?). If published, this will include your full peer review and any attached files.

Reviewer #1: No

Reviewer #2: No

Reviewer #3: No

---

## [Author Response · Author response to Decision Letter 1]

11 Sep 2020

Response to Reviewers: for manuscript #PONE-D-19-18201, “Computer-based inhibitory control training in children with attention-deficit/hyperactivity disorder (ADHD): Evidence for behavioral and neural impact” by Meyer, Santillana, Miller, Clapp, Way, Bridgmann-Goines, & Sheridan. 

Dear Dr. Fernandez,

Please find attached a revised version of this manuscript incorporating the feedback and suggestions from Reviewer 3. There were no further comments from Reviewer 1 and Reviewer 2. Below are our detailed responses to the points raised. The reviewer’s comments are shown in italics, and our replies are directly below each comment with reference to the sections edited. 

In addition to the specific comments by the reviewers, documented below, we read carefully the editors comments, 

“I consider several of the Reviewer3's suggestions to be important for improving the article. In particular: a) include in the Introduction the existing controversy regarding the absence versus presence of results when cognitive training is applied (maybe differences in treatment efficacy is related to training time, number of sessions or other variables that have not been made explicit in the document) and b) be very careful when making statements like "the first study of its kind to use computerized training of IC in children with ADHD"; Reviewer3 indicates that this is not the case, please carefully review the literature.”

Thank you. We have included further discussion of the existing controversy in the introduction (p5 – p6), in which we address the heterogeneity of findings and incorporate findings from meta-analyses of cognitive training. We have also carefully reviewed the document to ensure we do not claim to be the first study to train IC in children with ADHD. Additionally, we have added discussion of the studies suggested by Reviewer 3 in our section of the introduction reviewing previous work of computerized cognitive training in ADHD (p5 and p6).

Reviewer #3:

1. The authors might consider referencing the following related work and toning down the language related to the novelty of the present study and should consider referencing Shavlev, Tsal, & Mevorach (2002; they may have more recent work as well) and Jones, Katz, Buschkuehl, Jaeggi & Shah (2018). Specifically, in the abstract, the authors say that theirs is the only study in which cognitive training for ADHD has targeted inhibition. This is not the case.

REPLY: We appreciate this reviewer providing further examples of work that has utilized cognitive training to target inhibition, among other processes, in children with ADHD, and we have reviewed this work in included the suggested citations on pages 5 and 6. Additionally, we carefully reviewed the manuscript and have removed any claims of being the first study to target IC in children with ADHD.

2. In general, the authors do a good job describing their study and their analyses. However, most of the results are null. This is not a bad thing -- we should certainly publish such findings. However, the authors should be more upfront about this in the abstract, so as not to mislead the reader. Relatedly, the introduction section would do well to discuss that a large portion of cognitive training research has produced null or small effects. There are several metaanalyses showing both that cognitive training works and that it does not. I recommend that the authors acknowledge this ongoing debate and position their work as contributing to this ongoing discussion.

REPLY: Thank you. We agree that transparency of null or small effects are important, and have included a report of null effects in the abstract. Additionally, we now include discussion of the ongoing debate regarding the effect sizes and transfer of cognitive training in our introduction (pages 5 and 6), including reviews of relevant meta-analyses.

---

## [Editor Report · Decision Letter 2]

14 Oct 2020

Computer-based inhibitory control training in children with attention-deficit/hyperactivity disorder (ADHD): Evidence for behavioral and neural impact

PONE-D-19-18201R2

Dear Dr. Sheridan,

We’re pleased to inform you that your manuscript has been judged scientifically suitable for publication and will be formally accepted for publication once it meets all outstanding technical requirements.

Kind regards,

Thalia Fernandez, Ph.D.

Academic Editor

PLOS ONE
---

## [Editor Report · Acceptance letter]

16 Nov 2020

PONE-D-19-18201R2 

Computer-based inhibitory control training in children with attention-deficit/hyperactivity disorder (ADHD): Evidence for behavioral and neural impact. 

Dear Dr. Sheridan:

I'm pleased to inform you that your manuscript has been deemed suitable for publication in PLOS ONE. Congratulations! Your manuscript is now with our production department. 

Kind regards, 

on behalf of

Dr. Thalia Fernandez 

Academic Editor

PLOS ONE